# Morphological and Molecular Description of *Sarcocystis meriones* n. sp. from the Libyan Jird (*Meriones libycus*) in Kuwait

**DOI:** 10.3390/ani15172575

**Published:** 2025-09-02

**Authors:** Fatemah A. M. Aryan, Osama M. E. El-Azazy, Evelina Juozaitytė-Ngugu, Donatas Šneideris, Laila M. A. Tahrani, Dalius Butkauskas, Petras Prakas

**Affiliations:** 1Department of Science, College of Basic Education, Public Authority for Applied Education and Training, Kuwait City 23167, Kuwait; fa.aryan@paaet.edu.kw; 2Veterinary Laboratories, Public Authority of Agriculture Affairs and Fish Resources, P.O. Box 21422, Safat, Kuwait City 13075, Kuwait; el_azazy@hotmail.com (O.M.E.E.-A.); lailaazadi96@gmail.com (L.M.A.T.); 3State Scientific Research Institute Nature Research Centre, Akademijos Str. 2, 08412 Vilnius, Lithuania; evelina.ngugu@gamtc.lt (E.J.-N.); donatas.sneideris@gamtc.lt (D.Š.); dalius.butkauskas@gamtc.lt (D.B.)

**Keywords:** *Sarcocystis*, rodents, *Meriones libycus*, *Sarcocystis meriones*, ultrastructure, phylogeny

## Abstract

*Sarcocystis* spp. are heteroxenous protozoan parasites, which form sarcocysts in muscle tissue of intermediate hosts and sporocysts in the intestinal mucosa of definitive hosts. It has previously been discovered that rodents of the genus *Meriones*, commonly known as jirds or gerbils, can be infected by two *Sarcocystis* species, *S. hoarensis* and *S. dirumpens*. However, there is a lack of studies reporting on the role of rodents as intermediate hosts of these parasites in the Middle East. In the current study, a new species of *Sarcocystis*, named *Sarcocystis meriones*, was described in the Libyan jird (*Meriones libycus*) from Kuwait. Microscopic sarcocysts of the parasite were found in the thigh muscles of four of the jirds studied. The unique wall structure of *S*. *meriones* was determined to have protrusions resembling thuja or a cylinder and having lateral microvilli. DNA analysis showed the new species is closely related to similar parasites that infect mice and rats. The study suggests that predatory mammals preying on jirds likely serve as the definitive hosts for *S*. *meriones*.

## 1. Introduction

*Sarcocystis* parasites belonging to the order Apicomplexa are worldwide distributed protists having an obligatory two-host prey–predator life cycle [1]. Typically, sarcocysts develop in the muscles and central nervous system (CNS) of the intermediate host (IH), while oocysts sporulate in the small intestine of the definitive host (DH). More than 220 species of *Sarcocystis* spp. have been described in reptiles, birds, and mammals [1,2]. *Sarcocystis* species found in rodents are a highly diversified group of tissue cyst-forming coccidia, where the exact number of these parasites is not known. Rodents are natural or experimental IHs of over 40 *Sarcocystis* species [1,3]. Members of the family Muridae serve as IHs for half of the known *Sarcocystis* spp. found in rodents. Experimental infections have established that rodents of the genus *Meriones* can act as IHs for two *Sarcocystis* species, *S*. *hoarensis* and *S*. *dirumpens* [4,5,6,7]. Notably, no natural infections with *Sarcocystis* have been reported in *Meriones* spp. *Sarcocystis* species are generally more host-specific to their IHs than for their DHs [2,5,8]. For example, species such as *S*. *muris*, *S*. *muriviperae*, and *S*. *singaporensis* have been attempted to be transmitted to the Mongolian jird (*Meriones unguiculatus*), the Tristram’s jird (*Meriones tristrami*), and the Shaw’s jird (*Meriones shawi*), resulting in no success [5,9,10].

Rodents constitute 40% of mammals on the earth, with the identification of 220 species in 29 families, of which the family Muridae encompasses all small rodents, e.g., mice, rats, gerbils, and jirds [11]. Jirds of the genus *Meriones* are omnivores feeding on plants and insects [12]. In their habitats, they could be predators of desert insects, e.g., beetles and ants; on the other hand, they are preyed upon by wild mammals and reptiles. The Libyan jird (*Meriones libycus*) is distributed in North Africa and the Middle East and extends far east to Mongolia and China [13]. This wild rodent is widespread in the desert and steppes of northern and eastern Arabia, including Kuwait [14]. In the Middle East and North Africa, most previous studies have investigated rodents, including Libyan jirds, for their helminth biodiversity in Kuwait, Tunisia, and Iran [12,15,16]. As humans and grazing animals are portions of semi-desert and desert ecosystems, pathogens potentially carried by jirds (*Meriones* spp.) could be candidates for spillover to humans and livestock. For example, zoonotic helminths, e.g., *Hymenolepis nana* and *H*. *diminuta*, were reported in Libyan jirds and Sundevall’s jirds (*Meriones crassus*) in Kuwait [15] and *Capillaria hepatica* in the Persian jird (*Meriones persicus*) from Iran [17]. *Meriones* spp. were found to be reservoirs of some zoonotic protozoa, e.g., *Toxoplasma gondii* [18] and *Leishmania major* [19]. Various bacteria, such as *Anaplasma ovis*, *Rickettsia* spp., and *Bartonella* spp., were detected in Shaw’s jird [20], and *Anaplasma bovis* was identified in Libyan jird [21]. These pathogens could be a potential threat to humans and animals in wildlife.

The primary morphological diagnostic criterion for the description and differentiation of *Sarcocystis* species found in IHs is the structure of the sarcocyst wall, as examined by light and electron microscopy. Therefore, a detailed morphological characterization of sarcocysts is essential when describing a new *Sarcocystis* species [1]. Nevertheless, morphological analyses alone are insufficient to differentiate closely related *Sarcocystis* species that share similar structural traits. Nowadays, *Sarcocystis* spp. are described in IHs based on a combination of morphological and genetic characterization of the parasites. Sequencing of nuclear *18S* rRNA, *28S* rRNA, *ITS1*, and mitochondrial-encoded *cox1* is most widely used for the genetic analysis of *Sarcocystis* spp. [22]. It should be noted that there are no universal genetic loci for the discrimination of all species of the genus *Sarcocystis* [23]. Therefore, new nuclear, mitochondrial, and apicoplast DNA markers are increasingly suggested [24,25,26].

Very little work has been carried out on *Sarcocystis* infection in rodents in the Middle East. Using a digestion technique, Rahdar et al. [27] detected *Sarcocystis* bradyzoites in 50% of domestic mice (*Mus musculus*) and in 25% of brown rats (*Rattus norvegicus*), while Indian gerbils (*Tatera indica*) were free from infection. However, the *Sarcocystis* species was not identified in the study. More recently, *S*. *cymruensis* was found in the thigh muscles of 13 brown rats from Kuwait [24]. This study reports a new *Sarcocystis* species in Libyan jirds trapped from a semi-desert area in Kuwait based on light microscopy, ultrastructural analysis, and genetic characterisation of the parasite in four loci.

## 2. Materials and Methods

### 2.1. Sample Collection

The study area is a semi-desert region surrounding the Veterinary Laboratories (Coordinates: 29°17′49.62″ N, 47°46′45.09″ E), located in the Amghara district, Al-Jahra Province, Eastern Kuwait (Figure 1). The area contains edible trees, such as date palm (*Phoenix dactylifera*) and Christi’s thorn-jujube (*Ziziphus spina-christi*), and is bordered by a hedgerow of green buttonwood bushes (*Conocarpus erectus*). Some parts of the study area have been left uncultivated and are covered with natural herbs and grasses, e.g., saltlover (*Halogeton glomeratus*) and cheatgrass (*Bromus tectorum*). Beyond the study area lies an open, sandy desert.

As part of an epidemiological study of rodent parasites in Kuwait, Libyan jirds were trapped alive from the study area using 10 special steel wire traps (38 cm L × 23 cm W × 19 cm H). Traps were placed at selected sites at the entrance of their barrows and under trees in the evening and visited the next morning. Trapped rodents were placed in plastic bags and transferred to the necropsy laboratory of Public Authority of Agriculture and Fish Resources (PAAFR) within a few hours for analysis. Libyan jirds were identified according to [15,28]. The captured Libyan jirds were euthanized using chloroform (CHCl3). These rodents were treated carefully without stress in adherence to the guidelines issued by the Norwegian Committee for Research Ethics in Science and Technology [29]. This study was carried out according to the regulations set by PAAFR for conducting research work and with the approval of technical committee of the Animal Resources sector.

### 2.2. Morphological Examination

For the detection of sarcocysts of *Sarcocystis* spp., fragments of thigh muscle tissue (~1–3 mm) were pressed between two slides and examined under stereo- (Kern OZL-463 stereo microscope (Kern, Germany)) and light microscopes (Nikon ECLIPSE 80i light microscope with the INFINITY3 image analysis (Nikon Instruments Inc., New York, NY, USA, JAV)). Prevalence of *Sarcocystis* was evaluated in fresh-squashed muscle samples. In addition, pieces of muscle were fixed in 10% formalin prior to histopathological examination by embedding in paraffin, then sectioned at 4 µm thickness, stained with hematoxylin and eosin (H&E), and examined by the light microscope (LM).

For the detailed morphological and molecular analysis of observed sarcocysts, muscle samples of infected Libyan jirds were stored at −20 °C until delivery to the Laboratory of Molecular Ecology, State Scientific Research Institute Nature Research Centre, Vilnius, Lithuania. Sarcocysts were excised from fresh-squashed thigh muscles with the help of two preparation needles. These sarcocysts were morphologically characterized by LM, analysing the size and shape of the sarcocyst and describing the sarcocyst wall and bradyzoites located in the cyst. Transmission electron microscopy (TEM) analysis was performed as previously described in [24]. One sarcocyst isolated from the same Libyan jird was examined via TEM.

### 2.3. Molecular Analysis

Three sarcocysts isolated from three individual Libyan jirds were subjected to molecular examination. The isolation of genomic DNA was carried out from individual sarcocysts using GeneJet Genomic DNA Purification Kit (Thermo Fisher Scientific, Vilnius, Lithuania).

Using previously developed primers (Table 1), we attempted to amplify *18S* and *28S* ribosomal RNA (rRNA), internal transcribed spacers 1 and 2 (*ITS1* and *ITS2*), cytochrome c oxidase subunit I (*cox1*), cytochrome b (*cytb*), and apicoplast RNA polymerase beta subunit (*rpob*). Each PCR was carried out in a 25 μL volume consisting of 12.5 μL of Vazyme 2 × Taq Master Mix (Vazyme, Nanjing, China), 0.5 μM of each primer, 7.5 μL of nuclease-free water, and 4 μL of template DNA. The PCR was initiated with an initial hot start at 95 °C, followed by 40 cycles of 45 s at 94 °C, 60 s at 54–60 °C depending on the annealing temperature of primers, 70 s at 72 °C, and a final extension at 72 °C for 7 min. Positive (DNA of *S*. *rileyi*) and negative (nuclease-free water instead of DNA template) controls were used for each amplification with different primers. Visualization and evaluation of PCR products were conducted using 1% agarose gel electrophoresis. The amplified products were purified enzymatically with the help of ExoI and FastAP (Thermo Fisher Scientific Baltics, Vilnius, Lithuania).

Purified PCR products were subjected to directional Sanger sequencing using the same forward and reverse primers as for the PCR. Sequencing reactions were carried out using the BigDye^®^ Terminator v3.1 Cycle Sequencing Kit and the 3500 Genetic Analyzer (Applied Biosystems, Foster City, CA, USA) according to the manufacturer’s instructions. For the evaluation of interspecific genetic variation and selection of sequences for phylogenetic analyses, the obtained sequences were compared with those of various taxa representing the Sarcocystidae family using the Nucleotide BLAST 2.17.0 (https://blast.ncbi.nlm.nih.gov/, accessed on 5 August 2025) sequence similarity search algorithm [34].

### 2.4. Phylogenetic Analysis

Phylogenetic analyses were performed using MEGA11 (version 11.0.13) software [35]. Multiple sequence alignments of partial sequences of *18S* rRNA, *28S* rRNA, *ITS1*, and *cox1* were obtained using the MUSCLE algorithm. The final alignments of *18S* rRNA, *28S* rRNA, and *ITS1* contained 1666, 1498, and 887 nucleotide positions, including gaps, respectively, while the *cox1* alignment consisted of 1015 nucleotide positions. The reconstruction of phylogenetic relationships was carried out using the maximum likelihood method. Evolutionary models for each sequence data set were chosen based on the lowest values of the Bayesian Information Criterion. As a result, the Tamura-Nei + G + I, HKY + G + I, HKY + G, and GTR + G nucleotide substitution models were selected for *18S* rRNA, *28S* rRNA, *ITS1*, and *cox1*, respectively. Two species, *Neospora caninum* and *Toxoplasma gondii*, representing the Sarcocystidae family, were set as outgroups for *18S* rRNA, *28S* rRNA, and *cox1*. While *S*. *rileyi* was used as an outgroup for the highly variable *ITS1*. The bootstrap method with 1000 repetitions was used for testing the reliability of the phylogeny.

## 3. Results

### 3.1. Prevalence and Morphological Characterisation of Sarcocysts

*Sarcocystis* spp. sarcocysts were detected in the thigh muscles of 8.5% (4/47) of Libyan jirds from Kuwait. Sarcocysts detected in all four rodents were morphologically similar and most likely represented one species.

Sarcocysts detected in tight muscles of the Libyan jird were microscopic and filamentous with blunt ends and measured 1560 × 89 μm (900–2450 × 75–124 μm) (Figure 2A). In H&E-stained sections, insignificant pathological changes with very scanty leukocytic infiltration were observed. Additionally, in fresh squash preparations, the sarcocyst wall was thin and reached up to 1.5 μm in thickness (Figure 2B). Septa divided sarcocysts into compartments filled with lancet-shaped bradyzoites, 7.7 × 2.2 μm (6.1–9.0 × 1.2–3.0 μm) in size (Figure 2C). TEM analysis revealed that the villar protrusions (vp) of the sarcocyst wall have a proximal, sharply conical base and a distal segment that is thuja-like. The densely packed vp extending from the surface of the cyst were 1.2 × 0.5 µm in size, resembling a thuja or a cylinder and having lateral microvilli. The ground substance layer measured 0.5–0.6 μm in thickness (Figure 2D). The sarcocyst wall corresponds to type 22-like of the Dubey et al. [1] classification.

### 3.2. Molecular Characterisation of the Novel Sarcocystis Species

We were able to genetically characterise three isolates of *Sarcocystis meriones* within *18S* rRNA, *28S* rRNA, and *cox1* and one isolate of parasite species within *ITS1*. Whereas the amplification using primers targeting *rpoB*, *cytb*, and *ITS2* was unsuccessful. All three isolates of *S*. *meriones*, showed 100% identity within *18S* rRNA, *28S* rRNA, and *cox1*. Based on four genetic loci, sequences of *S*. *meriones* demonstrated the highest similarity with those of *S*. *ratti* from the black rat (*Rattus rattus*) and *S*. *myodes* from the bank vole (*Clethrionomys glareolus*) (Table 2). Furthermore, relatively high similarity was observed comparing our sequences of *S*. *meriones* with those of *Sarcocystis* sp. Rod1 isolated from two vole species, common vole (*Microtus arvalis*) and tundra vole (*Alexandromys oeconomus*), showing 99.2% similarity within *cox1* and 97.7% similarity within *28S* rRNA. However, the *cox1* and *28S* rRNA sequences of *Sarcocystis* sp. Rod1 available were relatively short, 619 bp and 735 bp in length, respectively. Therefore, these sequences of *Sarcocystis* sp. Rod1 were not included in further phylogenetic analysis.

Comparing the *18S* rRNA, *28S* rRNA, and *cox1* sequences of *S*. *meriones* with those of *S*. *ratti*, *S*. *myodes*, and *Sarcocystis* sp. Rod1, the obtained genetic similarity values were ≥97.7%. However, within *28S* rRNA and *cox1*, *S*. *meriones* showed relatively high differences from other *Sarcocystis* spp., exceeding 4.5%. When the 665 bp partial *ITS1* sequence of *S*. *meriones* was analysed using BLAST, significant genetic similarity was obtained only with sequences of *S*. *myodes* and *S*. *ratti*, and the values of genetic similarity were in the range of 78.2–78.7%.

The genetic variability between sequences of *S*. *meriones*, *S*. *myodes*, and *S*. *ratti* within *18S* rRNA, *28S* rRNA, *cox1*, and *ITS1* is provided in Table 3. At *cox1* these three species differed by 4–8 SNPs; at *18S* rRNA, and *28S* rRNA from seven to 28 SNPs and up to a single indel were determined between species pairs; and finally, at *ITS1* the number of indels (47–81) and SNPs (74–101) was very high. Thus, based on four analyzed genetic loci, *S*. *meriones* showed significant genetic differences from other *Sarcocystis* spp.

### 3.3. Phylogenetic Analysis of the New Sarcocystis Species

The phylogenetic analyses using four genetic loci (*18S* rRNA, *28S* rRNA, *ITS1*, and *cox1*) showed the placement of *S*. *meriones* with two *Sarcocystis* species, *S*. *myodes* and *S*. *ratti* (Figure 3). Based on all genetic loci examined, *S*. *meriones* was most closely related to *S*. *ratti*; however, significant clustering of these two species was supported by significant bootstrap values only in cases of *28S* rRNA and *cox1*. Furthermore, *S*. *meriones*, *S*. *myodes*, and *S*. *ratti* formed a sister clade to *S*. *cymruensis* and *S*. *muris* (Figure 3C). Other *Sarcocystis* species with rodents as IHs and birds (*S*. *jamaicensis* and *S*. *strixi*), opossums (*S*. *speeri*), or snakes (*S*. *muricoelognathis* and *S*. *pantherophisi*) as their DHs, and which were included in phylogenetic analyses, were placed in other clusters.

### 3.4. Description of Sarcocystis meriones n. sp.

The *Sarcocystis* parasite discovered in the present study was recovered from the tight muscles of the Libyan jird in Amghara district, Al-Jahra Province, Eastern Kuwait. The newly discovered parasite species was morphologically distinct by LM and TEM from *S*. *dirumpens* and *S*. *hoarensis*, found in rodents of the genus *Meriones* (Table 4). Based on TEM results, the sarcocyst wall of *S*. *meriones* was most similar to that of *S. villivillosi* found in muscles of laboratory rats and prior transmission of this *Sarcocystis* species to related rodent hosts was unsuccessful. However, villar protrusions of *S. villivillosi* were with a sharply conical base and a distal segment that is cocklebur-like with short, radiating projections, while the densely packed villar protrusions of *S*. *meriones* resembled thuja or a cylinder and had lateral microvilli (Table 4). The newly discovered parasite species was genetically characterised at four loci most commonly used for the differentiation of *Sarcocystis* spp. The comparison of sequences obtained in the present study and phylogenetic analysis revealed that *S*. *meriones* was genetically most similar to *S*. *ratti*, *S*. *myodes*, and *Sarcocystis* sp. Rod1, which form sarcocysts in the muscles of rodents. In comparison with the most closely related *Sarcocystis* species, small but significant genetic differences were observed in *cox1* (0.4–0.8%), *18S* rRNA (0.6–0.7%), and *28S* rRNA (1.1–2.3%). By contrast, the *ITS1* sequence of *S*. *meriones* showed less than 80% similarity with those of other *Sarcocystis* species, clearly confirming the genetic distinctiveness of the newly identified parasite. Based on results of TEM and molecular analysis, sarcocysts detected in thigh muscles of Libyan jirds from Kuwait were described for the first time and named as *Sarcocystis meriones* n. sp.
**Taxonomy summary of *Sarcocystis meriones* n. sp.****Type intermediate host:** Libyan jird (*Meriones libycus*).**Definitive host:** Unknown. Based on phylogenetic results, predatory mammals are suspected to serve as DHs.**Localit:y** Amghara district, Al-Jahra Province, Eastern Kuwait.**Morphology of sarcocysts:** By LM, sarcocysts were microscopic, filamentous with blunt ends, 1560 × 89 μm (900–2450 × 75–124 μm) in size with a thin (up to 1.5 μm) sarcocyst wall. Lancet-shaped bradyzoites measured 7.7 × 2.2 μm (6.1–9.0 × 1.2–3.0 μm). By TEM, densely packed protrusions of the sarcocyst wall were 1.2 × 0.5 µm in size, resembling a thuja or a cylinder and having lateral microvilli; the ground substance layer measured 0.5–0.6 μm; type 22-like**Genetic characteristics:** The new species was characterised by almost complete *18S* rRNA, partial *28S* rRNA, partial *ITS1*, and partial *cox1*. Obtained sequences showed ≤99.6%, ≤99.4%, ≤98.9%, and ≤78.7% genetic similarity to those of other *Sarcocystis* spp. within *cox1*, *18S* rRNA, *28S* rRNA and *ITS1*, respectively.**Type specimen:** Hapantotype 1 slide toluidine bluestained (NRCP00004) is deposited in the State Scientific Research Institute Nature Research Centre, Vilnius, Lithuania.**Sequences are available:** In the NCBI GenBank database under the accession numbers PV788216-PV788218 (*18S* rRNA), PV788219-PV788221 (*28S* rRNA), PV788222 (*ITS1*), and PV786926-PV786928 (*cox1*).**Etymology:** The Latin name of the genus *Meriones* is used for the species name.**ZooBank registration:** The Life Science Identifier (LSID) of the article is urn:lsid:zoobank.org:pub:1EE58E69-1A28-4C44-BDA6-3A45A46C5B16. The LSID for the new name Sarcocystis meriones is urn:lsid:zoobank.org:act:C12B8541-B611-4FAD-8372-A969D8105079.

## 4. Discussion

### 4.1. Morphological Differentiation of Sarcocystis meriones n. sp. from Other Sarcocystis Species

In the present work, sarcocysts detected in tight muscles of Libyan jirds were microscopic and filamentous with blunt ends and measured 900–2450 × 75–124 μm. The sarcocyst wall was thin and reached up to 1.5 μm in thickness under LM and bradyzoites were lancet-shaped and 6.1–9.0 × 1.2–3.0 μm in size. By TEM, the sarcocyst wall corresponds to type 22-like (Figure 2). Previously, cyst wall type 22 was identified exclusively for sarcocysts of *S*. *villivillosi* found in the muscles of the laboratory rat from the USA [1,37], and the DH of this *Sarcocystis* species is the boid snake (*Python reticulatus*) [37]. However, attempts to transmit *S*. *villivillosi* to other rodents, including jirds (*Apodemus agrarius*, *Apodemus flavicollis*, *Microtus arvalis*, *Gerbillus perpallidus*, and *Meriones unguiculatus*, etc.), were unsuccessful [5]. Furthermore, compared to *S*. *meriones*, *S*. *villivillosi* had striated sarcocyst walls and fusiform bradyzoites, 5 µm in length and 1 µm in width [37]. Moreover, the villar protrusions of *S*. *villivillosi* have a proximal, sharply conical base and a distal segment that is cocklebur-like, with short, radiating projections. Cross-sections of the distal segment of the projections resemble cogwheels. Meanwhile, the villar protrusions of *S*. *meriones* have a proximal, sharply conical base and a distal segment that resembles a thuja or a cylinder, with lateral microvilli (Table 4).

Previously, two *Sarcocystis* species, *S*. *hoarensis* and *S*. *dirumpens*, have been experimentally established in the muscle of the genus *Meriones* [5,6,7,36]. The macroscopically visible white sarcocysts of *S*. *hoarensis* were detected in the eyelids, the connective tissue of the snout, the tongue, the subcutis, and the connective tissue of the ears, as well as in the subcutis and connective tissue of the scrotum, anus, the base of the tail, and the soles of the feet of experimentally infected rodents [6]. Sarcocysts of *S*. *hoarensis* were seen with the naked eye and reached diameters of up to 2.5 mm. Meanwhile, the cyst wall of this species was smooth, and bradyzoites were approximately 11–12 µm in length and 2 µm in width [6]. Dubey et al. [1] assigned the wall of sarcocysts of *S*. *hoarensis* to type 42. Whereas *S*. *dirumpens* developed in the musculature of jirds experimentally exposed to sporocysts collected from a naturally infected butterfly viper (*Bitis nasicornis*) [5,6,36]. Sarcocysts of this species, two years post-infection, were macroscopic, measuring approximately 100–400 µm in width and up to 2.5 cm in length, depending on the muscles parasitized. The cyst wall of the septated sarcocysts appeared thin and smooth under LM; bradyzoites measured approximately 8–9 µm in length and 2 µm in width [6]. Dubey et al. [1] assigned the wall of sarcocysts of *S*. *dirumpens* to type 1b. Sarcocysts of *S*. *hoarensis* and *S*. *dirumpens* were found in jirds, gerbils, hamsters, and mice [6]. Thus, sarcocysts detected in our work differ significantly in morphology from those of *S*. *hoarensis* and *S*. *dirumpens* previously reported in rodents of the genus *Meriones*.

### 4.2. Potential Definitive Hosts of Sarcocystis meriones n. sp.

On the basis of *18S* rRNA, *28S* rRNA, *ITS1*, and *cox1*, *Sarcocystis meriones* n. sp. clustered in phylogenetic analyses with four *Sarcocystis* spp. (*S*. *cymruensis*, *S*. *muris*, *S*. *myodes*, and *S*. *ratti*) that employ rodents as IHs [24,38,39,40]. Specifically, *S*. *cymruensis* and *S*. *ratti* form sarcocysts in brown rats and black rats, respectively; *S*. *muris* infects domestic mice, and IHs of *S*. *myodes* are voles and mice of the genus *Apodemus* [38,40,41,42,43]. Domestic cats (*Felis catus*) have been shown to serve as DH of *S*. *cymruensis* and *S*. *muris* in transmission experiments [40,41]. The DHs of *S*. *myodes* and *S*. *ratti* have not yet been found, but phylogenetic evidence suggests that they could be predatory mammals [38,39]. The ecological conditions and landscape of the study area in Kuwait favor the thriving of Libyan jirds, as well as the presence of reptilian predators and wild carnivores, such as the desert monitor lizard (*Varanus griseus*), false cobra (*Malpolon moilensis*), and the red fox (*Vulpes vulpes*). As these predators coexist with Libyan birds, it is thought that one of them may be the DH of the new species. Based on phylogenetic results, the red fox could be suggested as DH of the new *Sarcocystis* species; however, this hypothesis should be confirmed by the transmission experiments. The present study may encourage further investigations of the new *Sarcocystis* species in other countries where Libyan jirds are present, to better understand its geographical distribution. Additionally, further research on other wild rodents in the semi-desert and desert regions of Kuwait is necessary to clarify the host specificity of *S*. *meriones*.

## 5. Conclusions

For the first time sarcocysts of *Sarcocystis* were detected in thigh muscles of the Libyan jird (*Meriones libycus*) collected from a semi-desert area in Eastern Kuwait. Based on microscopical (LM and TEM) and molecular examinations using four genetic loci (*18S* rRNA, *28S* rRNA, *ITS1*, and *cox1*), the new species *S. meriones* was described. The phylogenetic results of the present study suggest that predatory mammals are potential DHs of *S*. *meriones*. Future investigations are required to elucidate the host specificity, geographical distribution, and pathological effects of this newly described *Sarcocystis* species.

## Figures and Tables

**Figure 1 animals-15-02575-f001:**
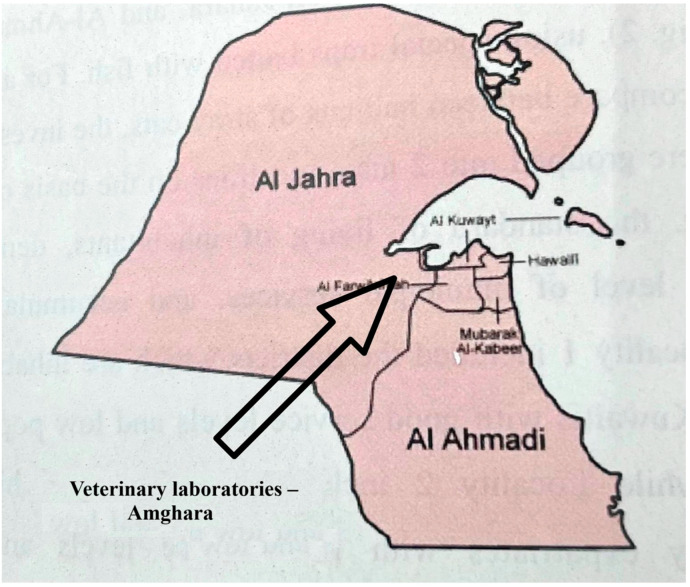
Location of the study area in the Amghara district, Al-Jahra Province, Eastern Kuwait. The area surrounds the veterinary laboratories (coordinates: 29°17′49.62″ N, 47°46′45.09″ E).

**Figure 2 animals-15-02575-f002:**
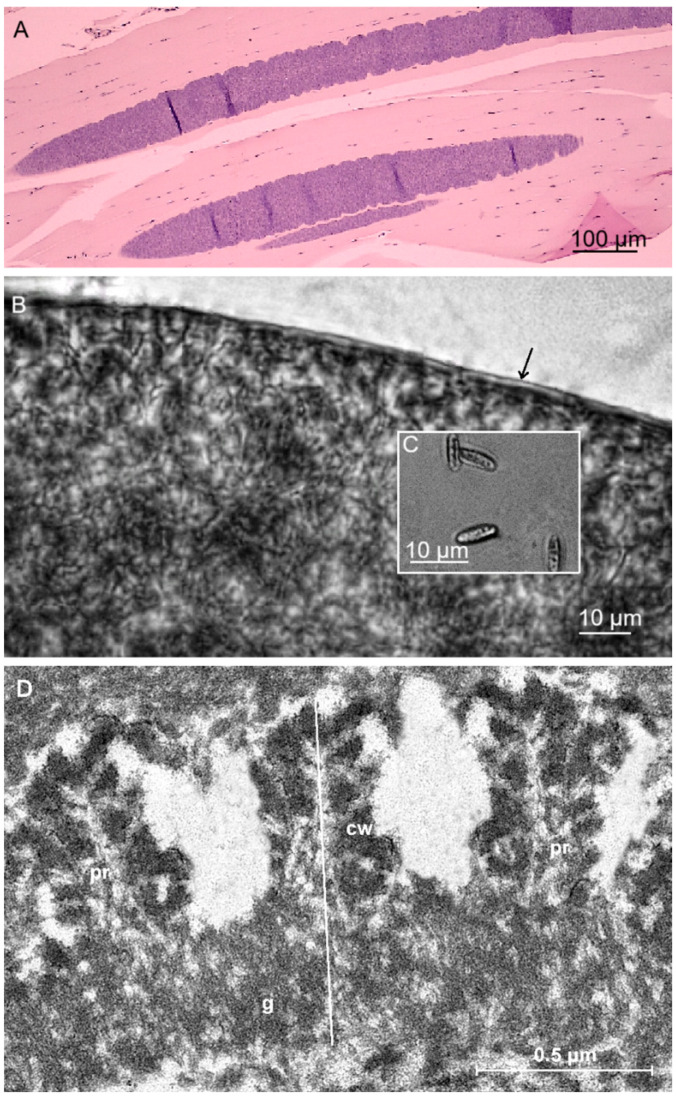
Morphology of *Sarcocystis meriones* n. sp. from muscle tissue of the Libyan jird (*Meriones libycus*) from Kuwait. (**A**) H&E-staining. LM micrograph showing spindle-shaped sarcocysts with thin cyst wall. (**B**,**C**) Light micrographs. Fresh preparations. (**B**) Fragment of sarcocyst. Note a thin and apparently smooth cyst wall (marked with arrow). (**C**) Lancet-shaped bradyzoites. (**D**) TEM micrograph; fragment of cyst wall (cw) with thuja or a cylinder protrusion and having lateral microvilli (pr); ground substance (g).

**Figure 3 animals-15-02575-f003:**
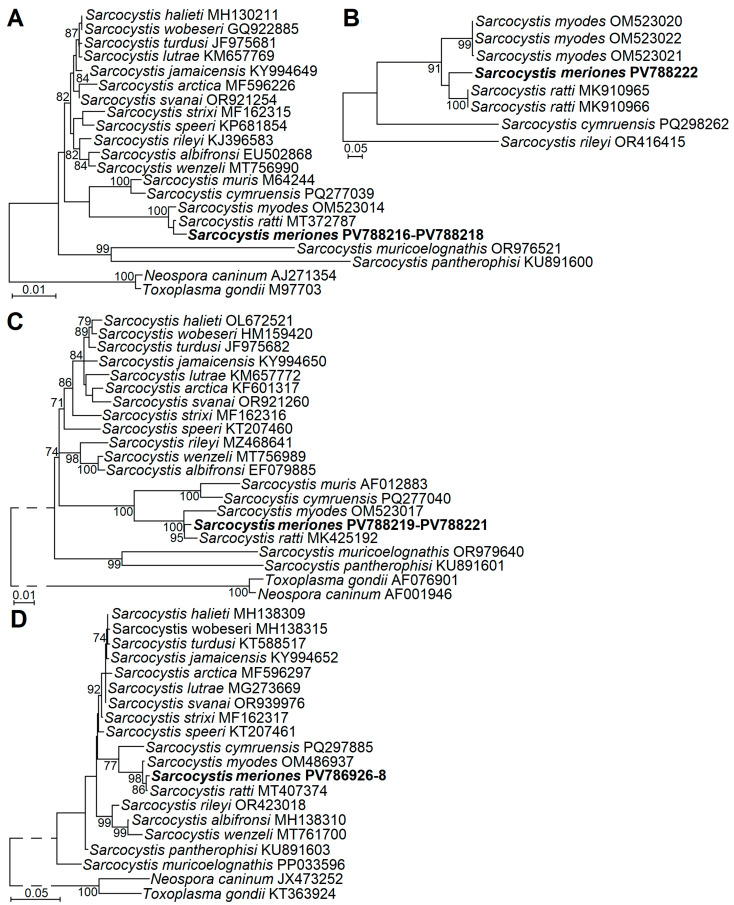
The phylogenetic placement of *Sarcocystis meriones* sp. n. from Lybian jird based on sequences of *18S* rRNA (**A**), *ITS1* (**B**), *28S* rRNA (**C**), and *cox1* (**D**). *Neospora caninum* and *Toxoplasma gondii* (**A**,**B**,**D**) or *Sarcocystis rileyi* (**C**) were used as outgroup species. Bootstrap values ≥ 70% are displayed next to branches.

**Table 1 animals-15-02575-t001:** The oligonucleotides used in this study and their characteristics.

Locus	Genome	Primer Name	Primer Sequence	Orientation	Product Length	Ref
*18S* rRNR	Nuclear	SUNIF1	ACCTGGTTGATCCTGCCAGT	Forward	~950	[24]
*18S* rRNR	Nuclear	SUNIR1	TTCGCAGTAGTTCGTCTTTAACA	Reverse	~950	[24]
*18S* rRNR	Nuclear	SUNIF3	GGGGCATTCGTATTTAACTG	Forward	~910	[24]
*18S* rRNR	Nuclear	SUNIR2	GATCCTTCCGCAGGTTCAC	Reverse	~910	[24]
*28S* rRNR	Nuclear	KL-P1F	TACCCGCTGAACTTAAGCAT	Forward	~1580	[30]
*28S* rRNR	Nuclear	KL-P2R	TGCTACTACCACCAAGATCTGC	Reverse	~1580	[30]
*ITS1*	Nuclear	SU1F	GATTGAGTGTTCCGGTGAATTATT	Forward	~1050	[31]
*ITS1*	Nuclear	5.8SR2	AAGGTGCCATTTGCGTTCAGAA	Reverse	~1050	[31]
*ITS2*	Nuclear	PSITS2F	GATGAAGGACGCAGTGAAATG	Forward	~1200	[24]
*ITS2*	Nuclear	PSITS2R	ATTTCCACTTTGAGCTCTTCC	Reverse	~1200	[24]
*Cox1*	Mitochondrial	SF1	ATGGCGTACAACAATCATAAAGAA	Forward	~1100	[32]
*Cox1*	Mitochondrial	SR5	TAGGTATCATGTAACGCAATATCCAT	Reverse	~1100	[32]
*Cytb*	Mitochondrial	80CYTBF2	ATGAGTTTAGTGCGAGCACATTT	Forward	~1080	[25]
*Cytb*	Mitochondrial	1080CYTBR2	TTAATATAGACATACAGCTAAGCTTGTGA	Reverse	~1080	[25]
*rpoB*	Apicoplast	RPOBF	TAGTACATTAGAAATCCCTAAAC	Forward	~850	[33]
*rpoB*	Apicoplast	RPOBR	TCWGTATAAGGTCCTGTAGTTC	Reverse	~850	[33]

**Table 2 animals-15-02575-t002:** The genetic comparison of *Sarcocystis meriones* n. sp. with other *Sarcocystis* species.

Region	Length	GenBank acc. no.	Highest Genetic Similarity Compared with Other *Sarcocystis* spp.
*18S* rRNA	1759	PV788216–PV788218	99.4% *S*. *ratti* (MT372787, MK425189-90), 99.3% *S*. *myodes* (OM523014-16), 98.0% *S*. *rileyi* (GU120092)
*28S* rRNA	1546	PV788219–PV788221	98.9% *S*. *ratti* (MK425192-93), 98.4% *S*. *myodes* (OM523017-19), 97.7% *Sarcocystis* sp. Rod1 (OQ557457-58), 94.1% *S*. *cymruensis* (PQ277040)
*Cox1*	1053	PV786926–PV786928	99.6% *S*. *ratti* (MK430072-73, MT407374), 99.2% *S*. *myodes* (OM486937-39), 99.2% *Sarcocystis* sp. Rod1 (OQ558008-09), 95.4% *S*. *cymruensis* (MT407373, PQ297885)
*ITS1*	665	PV788222	78.7% *S*. *ratti* (MK910965-66), 78.2–78.6% *S*. *myodes* (OM523020-22)

**Table 3 animals-15-02575-t003:** The genetic variability between sequences of *S*. *meriones*, *S*. *myodes*, and *S*. *ratti*.

Genetic Region	*S*. *meriones* vs. *S*. *ratti*	*S*. *meriones* vs. *S. myodes*	*S*. *myodes* vs. *S*. *ratti*
*18S* rRNA	1749/1759 (10 SNP)	1746/1758 (11 SNP, 1 indel)	1769/1777 (7 SNP, 1 indel)
*28S* rRNA	1460/1476 (15 SNP, 1 indel)	1521/1546 (24 SNP, 1 indel)	1447/1475 (28 SNP)
*Cox1*	1049/1053 (4 SNP)	1045/1053 (8 SNP)	1047/1053 (6 SNP)
*ITS1*	525/667 (74 SNP, 68 indels)	541–544/692 (101–104 SNP, 47 indels)	499/677 (97 SNP, 81 indels)

**Table 4 animals-15-02575-t004:** Morphological comparison of *S*. *meriones* n. sp. from the Libyan jird (*Meriones libycus*) with two *Sarcocystis* species previously detected in *Meriones* jirds and with *S*. *villivillosi,* which forms sarcocysts most similar in morphology to those described in the present study.

*Sarcocystis* spp.	Light Microscopy	Bradyzoites	Electron Microscopy
*S*. *dirumpens* [1,8,36]	Macroscopic, up to 25 mm long and 100–400 µm wide; the sarcocyst wall smooth, without visible protrusions	~8–9 µm long and 2 µm wide	Parasitophorous vacuolar membrane with pleomorphic blebs, conical, irregular. The diameter of the primary wall complex from the base of the invaginations to the distal ends of the evaginations measured 100–160 nm (60–125 d.p.i.) or 80–100 nm (138 and 418 d.p.i.). Ground substance— 0.5–1.0 µm. Type 1b
*S*. *hoarensis* [1,6,7]	Macroscopic, up to 2.5 mm; the sarcocyst wall smooth, without visible protrusions	11–12 µm long and 2 µm wide	Parasitophorous vacuolar membrane lined with thick electron-dense layer, numerous irregular villar protrusions; has secondary cyst wall. Type 42
*S*. *villivillosi* [1,37]	Spindle-shaped, up to 0.7–1.1 mm long and 70–100 µm wide; sarcocyst wall 1.5 μm thick, striated	Fusiform, 5 µm long and 1 µm wide	Villar protrusions with a proximal, sharply conical base and a distal segment that is cocklebur-like with short (1.6 × 0.5 µm), radiating projections. Cross sections of the distal segment of projections appear as cogwheels. The core contains numerous vesicles, especially in the distal half. Type 22
*S*. *meriones* n. sp.	Microscopic and filamentous with blunt ends, 1560 × 89 μm (900–2450 × 75–124 μm); the sarcocyst wall smooth, up to 1.5 μm	Lancet-shaped 7.7 × 2.2 μm (6.1–9.0 × 1.2–3.0 μm)	Villar protrusions of the sarcocyst wall have a proximal, sharply conical base and a distal segment that is thuja-like. The densely packed villar protrusions—1.2 × 0.5 µm, resembling thuja or a cylinder and having lateral microvilli. The ground substance layer—0.5–0.6 μm. Type 22-like

## Data Availability

The *18S* rRNA, *28S* rRNA, *cox1*, and *ITS1* sequences generated in the present study were submitted to the GenBank database under accession numbers PV786926–PV786928, PV788216–PV788222.

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
