# Peer review of "Morphological and Molecular Description of *Sarcocystis meriones* n. sp. from the Libyan Jird (*Meriones libycus*) in Kuwait"

_animals, 2025, doi:10.3390/ani15172575_

Round 1

Reviewer 1 Report

Comments and Suggestions for Authors

The manuscript proposes a new species, Sarcocystis meriones, based on morphological and molecular characterization from the intermediate host, the Libyan jird (Meriones libycus ). I support the erection of this new species. However, the following revisions are recommended to strengthen the manuscript:

Transmission Electron Microscopy (TEM) Figures:

Additional TEM images should be provided to clearly illustrate the distinguishing ultrastructural features of S. meriones. The current figures do not sufficiently demonstrate key characteristics. The resolution of Figure 2C and 2D needs improvement, particularly 2D, as the current image quality makes it difficult to confidently compare with S. villivillosi (Type 22). The description should also be refined to clarify similarities/differences.

Keywords (Line 38):

The keywords should include the proposed species name, "Sarcocystis meriones", while removing redundant terms like "Middle East" and "novel species."

Molecular Methods (Line 128):

The seven molecular markers (18S, 28S, ITS1, ITS2, cox1 , cytb , and rpoB ) were amplified, but the primer sequences used should be included either in the main text or as supplementary data for reproducibility.

TEM Sample Information (Line 121):

Clarify how many sarcocysts were examined via TEM and whether they were isolated from the same or different Libyan jirds. This is important for assessing intraspecific variation.

Author Response

Reviewer 1
Comments and Suggestions for Authors

The manuscript proposes a new species, Sarcocystis meriones, based on morphological and molecular characterization from the intermediate host, the Libyan jird (Meriones libycus). I support the erection of this new species. However, the following revisions are recommended to strengthen the manuscript:

Point 1. Transmission Electron Microscopy (TEM) Figures:

Additional TEM images should be provided to clearly illustrate the distinguishing ultrastructural features of S. meriones. The current figures do not sufficiently demonstrate key characteristics. The resolution of Figure 2C and 2D needs improvement, particularly 2D, as the current image quality makes it difficult to confidently compare with S. villivillosi (Type 22). The description should also be refined to clarify similarities/differences.

Response 1. Thank you for your valuable comment. We have replaced Figures 2C and 2D with higher-quality photos to better highlight the similarities and differences between S. meriones and S. villivillosi. Furthermore, we have provided a table (Table 4) that clearly compares the morphological features of the described species with those of closely related species.

Point 2. Keywords (Line 38): The keywords should include the proposed species name, "Sarcocystis meriones", while removing redundant terms like "Middle East" and "novel species."

Response 2. Thank you for your suggest. We deleted "Middle East" and "novel species", and added "Sarcocystis meriones".

Point 3. Molecular Methods (Line 128): The seven molecular markers (18S, 28S, ITS1, ITS2, cox1, cytb, and rpoB ) were amplified, but the primer sequences used should be included either in the main text or as supplementary data for reproducibility.

Response 3. Thank you for your comment. We agree and provided the primer list in Table 1.

Point 4. TEM Sample Information (Line 121): Clarify how many sarcocysts were examined via TEM and whether they were isolated from the same or different Libyan jirds. This is important for assessing intraspecific variation.

Response 4. Thank you for your meaningful comment. We added the missing information in lines 121-122: “One sarcocyst isolated from the same Libyan jird was examined via TEM”.

Reviewer 2 Report

Comments and Suggestions for Authors

The article should be considered for acceptance after addressing the following suggestions.

Comments on the Quality of English Language

Language should be improved, including typical scientific reporting. The author's used first part reporting even in instances where this could have been avoided. 

Author Response

Reviewer 2
Comments and Suggestions for Authors

The article should be considered for acceptance after addressing the following suggestions.

A brief summary

The aim of the study was to describe a new Sarcocystis species, Sarcocystis meriones n. sp., isolated from the thigh muscle of a Libyan jird. The study used appropriate methods, and sufficient genetic markers and primers to confirm morphological identification description. The results highly contributes towards the Sarcocystis species available utilising rodents, especially common species such as Meriones, as intermediate hosts, and the further elucidated potential definitive hosts based on the relationship of this newly described species with other Sarcocystis species and cladal clustering. Hence providing a starting point of assessing potential definitive hosts, experimentally. This is crucial as most Sarcocystidae species still have incomplete life cycle to date, with either intermediate or definitive hosts still not defined.

General concept comments

Point 1. While sufficient identification procedures have been followed, the authors could have provided more comparative morphological descriptive data with other species, in a tabular form. This would have provided a more clearer way of eliminating other species, but also showed some potential similarities between the species.

Response 1. Thank you. Thank you for your valuable comment. We have provided a table (Table 4) that clearly compares the morphological features of the described species with those of closely related species.

Point 2. Furthermore, the authors could have calculated the genetic distances between the species and within clades for all different markers.

Response 2. Thank you very much. We have calculated the exact differences between S. meriones and S. myodes, as well as the two most closely related species. Please refer to Tables 2 and 3. In these cases, the genetic distance between these species would mirror the data already provided.

Point 3. The phylogenetic tree figures can be improved.

Response 3. Thank you very much. We have improved the phylogenetic tree figure.

Point 4. The references formatting need closer attention; some references have publication years ‘bolded’, while others don’t have. Please provide the doi for all references.

Response 4. Thank you very much. We reviewed our references more closely, provided the DOI where available, and corrected any mistakes.

SPECIFIC COMMENTS

Simple summary

Point 5. Line 16: Rephrase the statement starting with ‘however’ to “However, there is a lack of studies reporting on the role of rodents as the intermediate hists of these parasites in the Middle East”

Response 5. Thank you very much. We changed as you suggested.

Point 6. Line 18: Replace “has been discovered” with “was described”

Response 6. Thank you for your meaningful comment. We changed.

Introduction

Point 7. Line 48-51: Clarity is needed in these statements. The author’s noted no natural infections of Sarcocystis spp. in Meriones spp. in line 51, however the preceding statement report on Meriones acting as IHs of Sarcocystis spp. If the statement on Line 48 was based on experimental infections, the author’s should clearly state that, and combine the two statements.

Response 7. Thank you for valuable comment. We clarified this statement to be clear: “Experimental infections have established that rodents of the genus Meriones can act as IHs for two Sarcocystis species, S. hoarensis and S. dirumpens”.

Point 8. Line 52: …..host-specific ‘to their”

Response 8. Thank you. We changed.

Materials and Methods

Point 9. Line 106: This study was “carried out” or “conducted”

Response 9. Thank you. We changed.

Point 10. Line 112: Which type of processing 10% formalin does? If its fixation, be specific that the muscles were fixed in 10% formalin prior histopathological “examination”.

Response 10. Thank you for your meaningful comment. We changed.

Results

Point 11. The author’s should consider providing a table, clearly comparing the morphological features of the described species and closely related species. This could provide the similarities and differences in the diagnostic features and measurements. Furthermore, the authors tend to report in first part, please refrain from this. The figures need improvement.

Response 11. Thank you for valuable comment. We made the Table and compared the morphological features of the described species and closely related species. We changed our report from first paragraph as you suggested. We have replaced Figures 2C and 2D with higher-quality photos to better highlight the similarities and differences between S. meriones and S. villivillosi.

Point 12. Line 164: Delete “ in sustained squashed muscles of four infected rodent”. The sentence could read “Sarcocysts detected in all four rodents were morphologically similar and most likely represented one species”

Response 12. Thank you. We deleted.

Point 13. Line 165: This statement id better fitted for the abstract, and/or 3.3

Response 13. Thank you for your thoughtful comment. We have moved it to subsection 3.3 as you suggested.

Point 14. Line 188: The authors should refrain from reporting in first part. See line 190.

Response 14. Thank you for your insightful comment. We changed.

Point 15. Line 195: The use of the term ‘furthermore’ would be more appropriate compared to ‘also’

Response 15. Thank you. We changed. 

Point 16. Line 199: replace ‘are’ with ‘were’

Response 16. We replaced. 

Point 17. Line 203: The statement needs restructuring. The authors could have easily indicated that “The 18S rRNA, 28S rRNA and cox1 sequences of S. meriones showed more genetic relatedness with S. ratti, S. myodes and Sarcocystis sp. Rod1, indicated by genetic similarities of ≥97.7%.”

Response 17. Thank you for your meaningful comment. We changed.

Point 18. Line 224: There is no need for referred information on the results. This should be moved to the discussion, with more context.

Response 18. Thank you for your insightful comment. We have deleted it.

Point 19. Line 238: “….was recovered from the tight muscles of the ….”

Response 19. We changed.

Point 20. Line 243: If the authors provided comparable table in the results section, the reference of the discussion in this section would not be necessary. Thus allowing an easy transition to discussion without overly redescribing the results.

Response 20. Thank you. We have provided a table (Table 4) that clearly compares the morphological features of the described species with those of closely related species.

Point 21. Line 247: The significant genetic differences eluded to here should be calculated. Include cladal and species within the same clade genetic distances. This is critical for cases where there is not cladal support value in the case of ITS1 or between the species as with 18S rRNA.

Response 21. Thank you for valuable comment. We have calculated it: “In comparison with the most closely related Sarcocystis species, small but significant genetic differences were observed in cox1 (0.4–0.8%), 18S rRNA (0.6–0.7%), and 28S rRNA (1.1–2.3%). By contrast, the ITS1 sequence of S. meriones showed less than 80% similarity with those of other Sarcocystis species, clearly confirming the genetic distinctiveness of the newly identified parasite”.

Discussion

Point 22. The author’s should remove subtitles from the discussion. Only then, the author’s will realise the first paragraph of the discussion have no link to their results, and can easily fit with the background as it provide the typical procedures for identification and classification of new species.

Response 22. Thank you. We moved our first paragraph to the backround as suggested. Based on our experience in publishing in MDPI branch journals, including Animals, dividing Discussion into sections 4.1., 4.2, … is generally accepted and useful for the readers.

Point 23. Line 302: Were these established experimentally? Indicate that if so.

Response 23. Thank you for your insightful comment. We added the missing information.

Comments on the Quality of English Language

Point 24. Language should be improved, including typical scientific reporting. The author's used first part reporting even in instances where this could have been avoided.

Response 24. Thank you. We improved English language using QuillBot program.